# A Protective Factor for Emotional and Behavioral Problems in Children: The Parental Humor

**DOI:** 10.3390/children9030404

**Published:** 2022-03-11

**Authors:** Benito León-del-Barco, Santiago Mendo-Lázaro, María-Isabel Polo-del-Río, Fernando Fajardo-Bullón, Víctor-María López-Ramos

**Affiliations:** 1Department of Psychology, Faculty of Teacher Training College, University of Extremadura, 10071 Cáceres, Spain; smendo@unex.es (S.M.-L.); mabelpdrio@unex.es (M.-I.P.-d.-R.); vmlopez@unex.es (V.-M.L.-R.); 2Department of Psychology, Education and Psychology Faculty, University of Extremadura, 06006 Badajoz, Spain; fernandofajardo@unex.es

**Keywords:** humor parental, parents, mental health, emotional disorders, behavioral disorders, children, adolescents

## Abstract

In order to prevent the development of emotional and behavioral problems, risk and protective factors must be identified. This study aims to establish a link between perceived parental humor and children mental health. The sample comprises 762 pupils aged 10 to 15 (M = 12.23; SD = 1.12), who completed self-reports evaluating perceived parental humor (*EEE-H*) and their own emotional and behavioral strengths and difficulties (*SDQ*). The results indicate that parental humor is negatively associated with internalizing problems in children; no association with externalizing problems is observed. According to the study, girls who perceive low parental humor are the most likely to display internalizing problems, while girls perceiving high parental humor are the least likely to do so. Parental humor, characterized by calmness, cheerfulness, and optimism, is a protective factor against internalizing problems in children, especially girls. We recommend training for parents and training and intervention programs for families to encourage activities that boost parental humor.

## 1. Introduction

Emotional and behavioral problems in children and adolescents are linked to anxiety, distress, disability, and functional impairment and place an economic burden on public health systems [1]. Preventing the development of emotional and behavioral problems requires risk and protective factors to be identified. The family plays a key role as a risk or protective factor against mental health problems [2]. Parents’ approaches to child rearing, interpersonal relationships, and parenting styles are key factors in children’s development. It is fundamental that we seek to understand individual differences in parent-child relationships, because the quality of these relationships influences in the development [3].

Interpersonal relationships, parenting styles, and parental opinions on education are factors associated to children’s development. Parenting styles have been used to understand the effects of family on children’s competences. Baumrind [4] has been particularly influential in the study of parenting styles, outlining the various styles which are now well-known in the scientific literature. Most studies follow the two-dimensional framework designed by Maccoby & Martin [5], which classifies parents into four different styles (authoritarian, authoritative, permissive, and neglectful) on the basis of two areas: warmth and control [6].

Other, less widely studied dimensions include autonomy support, psychological control [7,8], disclosure, and parental humor, which should be taken into consideration in a multidimensional model of parenting styles [9].

The concept of humor has been extensively researched and has grown in relevance in recent years as a result of its social, psychological, and physical benefits [10,11,12,13]. Four styles of humor have been identified, two of which are positive or adaptive: (1) affiliative humor, which promotes and facilitates cheerful, positive interpersonal relationships based on jokes; and (2) self-enhancing humor, which fosters a humorous approach to life, used as a mechanism by which to regulate emotions. The other two styles are negative or potentially harmful: (3) aggressive humor, which is the use of humor to criticize or manipulate others, and is socially inappropriate; and (4) self-defeating humor, which is excessively self-deprecating humor, laughing at oneself when ridiculed or belittled [14]. 

Scientific research has demonstrated the influence of humor on mental health, with a number of studies finding a link between positive humor and emotions leading to improved mental health (lower levels of depression and anxiety, higher self-esteem and positive emotions) [15]. Other research has focused on exploring personal strengths, including optimism, cheerfulness, and sense of humor, concluding that these strengths reduce negative emotions and boost positive emotions, making a significant impact on people’s physical and mental health and quality of life [16,17].

As research into the effects of humor has progressed, it has been demonstrated that humor on its own does not always have a positive impact. One of the types of humor considered to be potentially healthy and beneficial is affiliative humor [18], which serves to facilitate interpersonal relationships [19]. A positive correlation has also been observed between affiliative humor and parental warmth, as well as between aggressive and self-defeating humor and parental rejection [20].

For these reasons, we have focused on parental humor in this study, understood as a relational attitude adopted by parents that is characterized by calmness, cheerfulness, and optimism, and creates a family environment which encourages communication and promotes socialization among children [10,21]. If children learn to cope with difficult situations using humor, this will help them to withstand negative emotions such as sadness, disappointment and distress [22].

### The Present Study: Humor Parental and Emotional and Behavioral Problems

Parental humor is one of the dimensions characterizing parent–child relationships, creating a more supportive family environment and promoting children’s wellbeing [10,21]. The use of humor within the family helps to maintain a healthy relationship between family members [22].

What is the impact of humor on children? According to a number of studies, parental humor, as well as other dimensions of parenting styles such as warmth/communication, autonomy, behavioral control, and disclosure, has an impact on adolescents’ emotional stability [23]. An environment in which humor prevails, characterized by optimism, cheerfulness, calmness, and laughter, can provide an appropriate framework for expressing emotions and withstanding negative emotions [19,24].

Research has demonstrated that children’s social and intellectual development can be improved and enriched when they are exposed to a regular, structured, and appropriate use of humor by their parents, emphasizing the value and benefits of humor as an educational tool in the home. The use of humor under these conditions can teach appropriate moral and civic behavior, boost critical thinking skills, promote values, and generate a sense of responsibility in children and adolescents [22]. Some studies observe that a sense of humor in children helps them to build resilience and cope with difficult situations [25,26].

Various authors have studied the relationships between different dimensions of parenting styles and adolescent adjustment in terms of the internalizing and externalizing problems which may be displayed, finding an association between warmth, autonomy support, disclosure, and positive adjustment among adolescents and a reduction in externalizing problems and psychopathological symptoms [10,27,28]. 

Likewise, these investigations have shown that psychological control is associated with internalizing symptoms and behavioral control predicts externalizing symptoms. Parental behavioral dimensions, such as psychological and behavioral control, are the most studied. In this work we intend to expand the parental dimensions and study the role of parental humor (protective factor) in internalizing and externalizing symptomatology. Parental humor is one dimension of parenting styles which is considered to be linked to improved internal and external adjustment in children and adolescents [27]. More specifically, several studies conclude that optimism and the presence of humor in the family communication pattern prevents psychopathological intensity [23].

Our primary objective in this study is to establish the association between parental humor as perceived by children and adolescents and their mental health. In order to do this, after analyzing the role of gender and age in parental humor and in internalizing and externalizing problems using binomial logistic regression analysis, odds ratio, and decision trees, the probability of displaying internalizing and externalizing problems was determined in relation to parental humor (low, medium, and high) as perceived by children, gender, and age. Our hypothesis is that high parental humor, characterized by calmness, cheerfulness, and optimism within the family, is a protective factor against the appearance of emotional and behavioral problems.

## 2. Materials and Methods

### 2.1. Participants

The sample was selected by means of multi-stage cluster sampling and random selection of groups in schools with several groups in Years 5 and 6 at primary level and Years 1 and 2 at secondary level. The cluster sampling was carried out by randomly selecting four schools. The number of participants (762) was determined on the basis of the number of pupils registered at primary and secondary level in public and state-funded private schools in Extremadura (Spain) during the 2018–2019 school year, considering a sampling error of 3% and a confidence level of 95.5%.

### 2.2. Instruments

#### 2.2.1. Strengths and Difficulties Questionnaire, SDQ

The self-report SDQ [29] is a brief instrument which is excellent for screening for mental health problems in children, with an acceptable internal consistency on all scales in both its international version [30,31] and its Spanish version [32,33]. It comprises 25 items divided into five dimensions or subscales (1. Emotional Symptoms, 2. Behavioral Problems, 3. Peer Relationship Problems, 4. Hyperactivity and 5. Prosocial Behavior). Each of the subscales is assessed via five items. A three-point Likert scale is used to record responses: 0 = No, not at all, 1 = Sometimes, and 2 = Yes, always. For community samples, it is recommended that the items from the Behavioral Problems subscale and the Hyperactivity subscale are grouped together in a new scale entitled Externalizing [34]. Equally, the items from the Emotional Symptoms subscale are added to the items from the Peer Relationship Problems subscale to create the Internalizing Problems scale) [35].

Both the total score from the SDQ and the scores from the various subscales and scales are classified into three categories: Normal, Borderline, and Abnormal. In the original scale [29], the limit of the Abnormal category corresponds to the score that delimits the top 10% of cases (Percentile ≥ 90), while the Borderline category corresponds to 10% of cases between the 80th percentile and 90th percentile. The total difficulties score is obtained by adding the four subscales. The Externalizing Problems scale obtains a Cronbach’s alpha (α) of 0.71 and a composite reliability (CR) of 0.76, while the Internalizing Problems scale obtains an α of 0.73 and a CR of 0.72.

#### 2.2.2. Scale to Evaluate the Parenting Style of Mothers and Fathers of Adolescents, Children’s Version (EEE-H)

The EEE-H [27] comprises 41 items grouped into six factors (Warmth/Communication, Autonomy Support, Behavioral Control, Psychological Control, Disclosure, and Humor), which are presented as a six-point Likert scale ranging from Totally disagree to Totally agree. In this study, we worked solely with the Parental Humor factor, comprising six items, which evaluates children’s perception of their parents’ sense of humor and optimism: “They’re almost always cheerful and optimistic”, “They’re usually in a good mood”, “They usually joke around with me”, “They’re usually calm and relaxed”, “They laugh a lot with me”, and “It’s fun doing things with them”. 

In the present study, reliability indexes for parental humor factor were the following: Cronbach’s alpha (α = 0.85), composite reliability (CR = 0.91), McDonald’s omega (Ω = 0.85), and average variance extracted (AVE = 0.63). In order to determine whether the factorial model found in the original validation study adequately adjusted to data, a confirmatory analysis was performed. As it can be observed, the model presents adjustment indexes showing evidence of reliability and validity for the generalization of results (χ2 = 102.931; χ2/df = 4.678; GFI = 0.957; IFI = 0.947; TLI = 0.912; CFI = 0.947; RMSR = 0.039; RMSEA = 0.077).

Why are we interested in children’s perception of their parents’ humor? There are low levels of convergence between parents’ opinions of their own parenting practices and those of their children. Parents’ own perceptions of their parenting practices can, at times, be biased by social desirability concerns. In this study, the perception of adolescent offspring is less biased and can be a more important predictor of their responses than their parents’ perception [27].

### 2.3. Procedure

This study has a cross-sectional, quantitative predictive design. We followed the ethical guidelines established by the American Psychological Association [36] with regard to informed consent from parents, as all participants were underage. Firstly, we contacted the schools to explain the objectives of the study and request authorization to complete the questionnaires. Then, we administered the questionnaires by class group. In this way, we were able to guarantee the anonymity of the responses, the confidentiality of the data obtained, and the exclusive use of this data for research purposes. The questionnaires were administered during school hours; the process took around 20 min in an appropriate setting without distractions. This study was approved by the Bioethics and Biosafety Committee of the University of Extremadura.

### 2.4. Data Analysis

First of all, reliability analyses were performed on the instruments. Multivariate analyses of the mean Parental Humor and Emotional and Behavioral Problems scores were carried out according to gender and age. Then, a binomial logistic regression analysis and odds ratio measure of association (OR) were performed. Finally, a classification model based on the decision tree statistical technique and using the CHAID division method (Chi-Squared Automatic Interaction Detection) was created to identify groups, reveal relationships between groups and predict future events. The statistical analysis was completed using the SPSS 21.0 package for PC.

## 3. Results

### 3.1. Demographic and Educational Characteristics of the Participants

The sample comprised 762 primary and secondary school students. The average age was 12.23 years old (SD = 1.122; range 10–15); 53.8% (*n =* 410) were female and 46.2% (*n =* 352) were male. With regard to the distribution of the participants by year group, 22% were in Year 5 at primary level, 23.1% in Year 6 at primary level, 26.3% in Year 1 at secondary level, and 28.6% in Year 2 at secondary level. 

### 3.2. Role of Gender and Age within the Variables Studied: Multi-Factor Analysis

In order to analyze the role of age and gender within the variables studied, multi-varied comparisons (MANOVA) of average results (Table 1), using as continuous response/dependent variables the scores of the Internalizing Problems and Externalizing Problems scales, as well as the score of the Parental Humor factor of the EEE-H, and as categorical predictor variables gender, age (≤12 years or ≥13 years), and gender/age interaction. 

Multivariate analysis of variance (MANOVA) revealed a significant main effect of gender (Wilks λ = 0.951, *F*(3, 756) = 13.001, *p* < 0.001, *ƞ* = 0.049), age (Wilks λ = 0.945, *F*(3, 756) = 14.684, *p* < 0.001, *ƞ* = 0.055), and gender/age interaction (Wilks λ = 0.983, *F*(3, 756) = 4.292, *p* = 0.005, *ƞ* = 0.017).

As for SDQ scales, univariate contrasts indicate that boys obtain higher scores in Externalizing Problems variable (*F*(1, 756) = 25.47, *p* < 0.001, *ƞ* = 0.034). Furthermore, ≤12 year-old participants get higher scores in Externalizing Problems variable (*F*(1, 756) = 3.934, *p* = 0.048, *ƞ* = 0.005) and in Internalizing Problems variable (*F*(1, 756) = 12.898, *p* < 0.001, *ƞ* = 0.017). There is no gender/age interaction whatsoever. As for the Parental Humor variable, univariate contrasts indicate that girls score higher than boys (*F*(1, 756) = 12.898, *p* < 0.001, *ƞ* = 0.013), and ≥13 year-old participants also get higher scores (*F*(1, 756) = 35.003, *p* < 0.001, *ƞ* = 0.044). There are no significant differences in gender/age interaction. 

### 3.3. Parental Humor and Emotional and Behavioral Problems: Binomial Logistic Regression Analysis and Odds Ratio (OR)

Our main goal is to determine the association between Parental Humor and mental health. A binomial logistic regression analysis has been carried out using two dependent variables: Internalizing Problems (emotional problems) scale scores and Externalizing Problems (behavioral problems) scale scores. These variables were turned into dichotomous variables at percentile 80 (*p* ≥ 80 = “*Suffering from a disorder”*; *p* < 80 = ”*Not suffering from any disorder”*). Subjects belonging to “*Abnormal”* and “*Adjacent”* categories from original scale variable (Peacock-Villada, DeCelles & Banda 2007) [26] were grouped within “Suffering from Internalizing Problems and Suffering from Externalizing Problems”. EEE-H factor *Parental Humor* was used as an independent variable, grouped as a trichotomous variable (Low Parental Humor = 1; Average Parental Humor = 2; High Parental Humor = 3) at the following percentiles (*p* ≤ 20 = 1; *p* > 20 y > 80 = 2; *p* ≥ 80 = 3). Gender and age of participants were included as control variables.

Table 2 shows the binomial logistic regression analysis with dependent variable Internalizing Problems (emotional problems). This regression analysis shows a satisfactory adjustment (χ2 = 32.173(7), *p* < 0.001; R Nagelkerke = 0.072), which allows a correct classification for 79.2% of the cases.

A thorough analysis of the results shows an association between parental humor and mental health emotional problems. The estimates of parameters reveal that Low Parental Humor (Wald = 23.732, *p* < 0.001) and Average Parental Humor (Wald = 9.257, *p* = 0.002) are significantly and directly associated with the Suffering from Internalizing Problems category. OR estimates show: (1) that the odds of having internalizing problems are 4.807x higher in the Low Parental Humor group than in the High Parental Humor; and (2) the odds of having internalizing problems are 2.405x higher in the Average Parental Humor group than in the High Parental Humor.

Table 3 shows the binomial logistic analysis with dependent variable Externalizing Problems (behavioral problems). This regression analysis does not show a satisfactory adjustment (χ2 = 3, 667(7), *p* = 0.817; R Nagelkerke = 0.064). The estimates of parameters reveal that Low Parental Humor (Wald = 0.175, *p* = 0.675) and Average Parental Humor (Wald < 0.001, *p* = 0.998) are not significantly associated with the Suffering from Externalizing Problems category.

### 3.4. Interpretation of Associations Observed and Role of Gender and Age: Classification Tree

Finally, with intent to ease the interpretation of significant associations observed and the role of both gender and age, a classification tree was made by dependent variable Internalizing Problems, adding independent variables Parental Humor and age and gender groups.

The tree correctly classifies 79.2% (Risk = 0.208; SE = 0.015) of participants. Figure 1 shows that girls with low perception of Parental Humor (Node 8) have the highest odds of suffering from Internalizing Problems (46.7%), while girls with high perception of Parental Humor (Node 6) have the lowest odds of suffering from Internalizing Problems (5.7%). Furthermore, within the group of participants with an average perception of Parental Humor, boys under 12 years old (Node 11) have the lowest probability of suffering from Internalizing Problems (32.6%).

## 4. Discussion

Regarding comparisons between scores on the SDQ scales according to gender and age, our results are consistent with prior research which confirms that boys obtain higher scores in Behavioral Problems and Hyperactivity [37], and that Externalizing and Internalizing Problems are more widely present in the early stages of adolescence [2,30,37,38]

In terms of the main objective of the study, which was to determine the association between parental humor and emotional and behavioral problems (mental health), the results show that a perception of high parental humor is a protective factor against internalizing problems in young people (emotional difficulties and peer relationship problems). This corroborates that parental humor is a factor that promotes healthy parenting associated with confident children and adolescents who search and enjoy healthy relationships with their peers [19]. Low or average parental humor scores are linked to a higher probability of developing internalizing problems, 4.807 times higher in the case of low parental humor and 2.405 for average parental humor. These results corroborate those of previous studies that stress the benefits of humor [10,11,12,13], and which demonstrate how important it is to study the influence of parenting styles on children’s mental health from a holistic perspective encompassing basic factors such as parental rejection as well as other relevant factors such as humor) [27,39,40].

Other studies with Spanish children are consistent with our results, demonstrating a negative correlation between parental humor and internalizing, depressive, and anxious symptomatology [28]. These studies support the conclusion that parental humor can act as a protective factor against the emergence of mental pathologies [23]. Parental humor, understood by children and adolescents as a display of optimism, good humor, and calmness by their parents, can boost their socialization [10] and improve peer relationships for adolescents with antisocial symptomatology [39]. Once again, these studies support our conclusion that high parental humor is a protective factor against peer relationship difficulties.

Why is parental humor associated with internalizing problems, especially emotional difficulties, and not with externalizing problems? In answer to this question, parental humor contributes to a warm parenting style [27], which, in turn, helps to create a positive family environment and reduce the presence of internalizing problems. This may be due to the fact that humor helps to regulate emotions [19,22,24] and is closely correlated with parental warmth and communication [14,39], serving as a tool to manage internalizing problems relating to emotions and peer relationships [41].

Why does this association have a greater impact on girls, protecting them from internalizing problems where there is a perception of high parental humor and increasing their vulnerability to internalizing problems where there is a perception of low parental humor? Regarding this second question, although men and women broadly share the same response strategy to humor, activation of the left prefrontal cortex, used in attention, is greater in women than in men [42], and tasks requiring greater emotional processing tend to generate more extensive activation of the limbic system in women [43]. In this regard, various studies have noted that women pay greater perceived attention to their emotions than [44,45,46], and, in general, women integrate emotional aspects into the various cognitive processes to a greater extent than men [47]. This would partially explain why the girls in our study obtained higher parental humor scores than the boys and why parental humor had a greater impact on emotional problems (internalizing problems) in girls.

Parental humor teaches children to remain optimistic despite their worries and difficulties and helps them to withstand negative emotions. The presence of humor in the family communication pattern prevents psychopathological intensity [23]. Moreover, parental humor enables effective communication between parents and children. Parental humor teaches children strategies and skills to develop assertiveness and interact in a positive way with others in different settings (family, school, friends, or sports). 

Finally, according to the fact that a more favorable family environment promotes the children’s well-being [10,21] and that the quality of the relationship between parents and children determines adolescents’ development [3], we believe that training aimed at parents can help them to understand that their children’s problems may originate in their own behavior. We recommend the use of training and intervention programs aimed at families. Parental training forms part of childrearing and is an effective method for promoting children’s development, improving parent–child relationships, facilitating assertive and effective communication with children, learning conflict resolution tools, and increasing parents’ sense of satisfaction and competence with regard to their tasks and responsibilities as mothers and fathers. Hence, improving children’s mental health and increasing the cost-effectiveness of any treatment required, as well as reducing overall costs [1]. To promote activities that foster parental humor as key dimension of a warm and effusive parental style could be considered as opposed to critical and lackadaisical styles causing absence of confidence or interiorized shame.

### Limitations and Future Research

This investigation has some limitations, such as the employment of self-reports by the children and adolescents. It would be appropriate to assess mental health and parental humor from the perspective of the parents also. Moreover, although maternal and paternal parenting styles tend to be highly consistent [27,48], the use of self-reports for both parents limits the identification and monitoring of potential differences between mothers and fathers. Finally, cultural influence means that the results must be interpreted with precaution when extrapolating them to other contexts [49].

## 5. Conclusions

In conclusion, this article demonstrates the link between parental humor and internalizing symptomatology. Parental humor emerges as a protective factor against emotional difficulties and peer relationship problems. For this reason, early monitoring and identification of variables such as perceived parental humor is very important [50], especially because children mimic parental humor from their very first months of life [51].

Equally, in order to prevent internalizing symptoms among children, it is recommended that parental humor is taken into consideration, especially among girls. More generally, humor helps us to tackle problems with an optimistic mindset and to establish positive relationships with others [22,25,26,47].

## Figures and Tables

**Figure 1 children-09-00404-f001:**
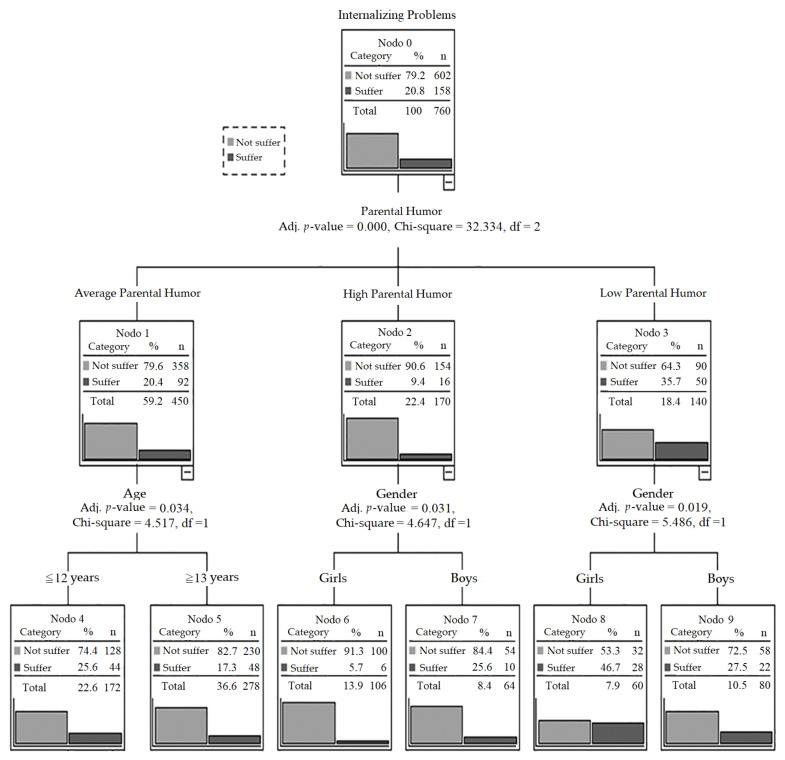
Internalizing Problems Classification Tree.

**Table 1 children-09-00404-t001:** Descriptors for the different variables studied. Internalizing Problems, Externalizing Problems, and Parental Humor by age and gender groups.

Variables	Gender	Age ≤ 12		Age ≥ 13		Total	
		M	SD	M	SD	M	SD
Internalizing Problems	Girls	8.02	2.94	6.77	2.83	7.23	2.93
	Boys	7.66	3.08	7.35	2.63	7.50	2.85
	Total	7.83	3.02	7.01	2.76	7.35	2.89
Externalizing Problems	Girls	8.07	2.66	7.98	2.21	8.02	2.39
	Boys	9.40	3.19	8.69	2.71	9.03	2.96
	Total	8.78	3.03	8.27	2.45	8.49	2.71
Parental humor	Girls	30.26	6.11	32.95	3.55	31.98	4.80
	Boys	29.56	6.02	31.28	4.81	30.47	5.47
	Total	29.89	6.06	32.14	4.20	31.28	5.17

**Table 2 children-09-00404-t002:** Binomial Logistic Regression and Odds Ratio based on dependent variable Internalizing Problems and Parental Humor.

	Internalizing Problems ^1^ (Emotional Problems)						
	B	SE	Wald	Sig.	OR	IC 95%	
Low Parental Humor ^2^	1.570	0.322	23.732	0.000	4.807	2.556	9.041
Average Parental Humor ^2^	0.877	0.288	9.257	0.002	2.405	1.366	4.232
Boys ^3^	0.073	0.185	0.156	0.693	1.076	0.749	1.545
≤12 years old ^4^	0.320	0.188	2.904	0.088	1.377	0.953	1.989

Reference category: ^1^ Suffering from Internalizing Problems. Comparison groups: ^2^ High Parental Humor; ^3^ Girls; ^4^ ≥13 years old.

**Table 3 children-09-00404-t003:** Binomial Logistic Regression and Odds Ratio based on dependent variable Externalizing Problems and Parental Humor.

	Externalizing Problems ^1^ (Behavioral Problems)						
	B	ST	Wald	Sig.	OR	IC 95%	
Low Parental Humor ^2^	0.118	0.282	0.175	0.675	1.125	0.647	1.956
Average Parental Humor ^2^	0.003	0.228	0.000	0.998	1.003	0.641	1.570
Boys ^3^	0.861	0.182	22.354	0.000	2.365	1.655	3.379
≤12 years old ^4^	0.419	0.182	5.283	0.022	1.521	1.064	2.174

Reference category: ^1^ Suffering from Externalizing problems. Comparison groups: ^2^ High Parental Humor; ^3^ Girls; ^4^ ≥ 13 years old.

## Data Availability

Data is contained within the article.

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
