# Peer review of "A Protective Factor for Emotional and Behavioral Problems in Children: The Parental Humor"

_children, 2022, doi:10.3390/children9030404_

Round 1

Reviewer 1 Report

This paper examined the relationship between parent humor and adolescent mental health. The paper is well designed and written. Here are some suggestions for the authors to consider in order to improve the manuscript:

1. The research gaps should be strengthened. 
The authors reviewed the literature on the relationships between parents' use of humor and adolescent development. However, research gaps are not clearly presented in the manuscript.

2. Line 168
The authors argued that adolescents' perceptions of parents' use of humor are more "objective" as compared to parents' own evaluations. While I agree with the adoption of adolescents' perception as it is closely linked to research questions (i.e., adolescent mental health), I recommend that the authors consider removing "objective" as adolescents' perceptions are also subjective. 

Author Response

Dear Reviewer:

We would like to thank you for considering our manuscript and for your careful review. We are pleased to submit a revised version of our article, which has been substantially improved as a result of your comments and suggestions. We have tried to modify every aspect that has been pointed out (see changes in red). We hope that you will consider this new version acceptable for publication.

 The authors appreciate all the recommendations made to improve the final manuscript.  We remain at your disposal for any questions you may have.

Sincerely,

The authors.

The authors reviewed the literature on the relationships between parents' use of humor and adolescent development. However, research gaps are not clearly presented in the manuscript.

RESPONSE: This has been done. New content has been included in  “The present study…”  section.

The authors argued that adolescents' perceptions of parents' use of humor are more "objective" as compared to parents' own evaluations. While I agree with the adoption of adolescents' perception as it is closely linked to research questions (i.e., adolescent mental health), I recommend that the authors consider removing "objective" as adolescents' perceptions are also subjective.

RESPONSE: This has been corrected.

Reviewer 2 Report

Methods:

The distribution of the sample under participants shift to the results section. 

Explain the detaisl of Multivariate analysis of variance used in the study for comapryiosns . 

Discussion:

The use of the parantal humor and clincial significanc eis missing.

The conclusion is too informative  and authors are requested to minimise the coclusion and move the content to discussion.

Always teh conclusion shoudl be objective based.

Author Response

Dear Reviewer:

We would like to thank you for considering our manuscript and for your careful review. We are pleased to submit a revised version of our article, which has been substantially improved as a result of your comments and suggestions. We have tried to modify every aspect that has been pointed out (see changes in red). We hope that you will consider this new version acceptable for publication.

 The authors appreciate all the recommendations made to improve the final manuscript.  We remain at your disposal for any questions you may have.

Sincerely,

The authors.

Methods:

The distribution of the sample under participants shift to the results section.

RESPONSE: This has been done.

Explain the detaisl of Multivariate analysis of variance used in the study for comapryiosns.

RESPONSE: This has been done.

Discussion:

The use of the parantal humor and clincial significanc eis missing.

The conclusion is too informative and authors are requested to minimise the coclusion and move the content to discussion.

Always teh conclusion shoudl be objective based.

RESPONSE: This suggestions are welcomed and have been modified.

Round 2

Reviewer 2 Report

Dear Authors

Thank you for your resubmission.

All the quires have been addressed satisfactorily.

All the best